

# Metagenomic next-generation sequencing for the clinical diagnosis and prognosis of acute respiratory distress syndrome caused by severe pneumonia: a retrospective study

Peng Zhang[1,2], Yan Chen[3,4], Shuyun Li[5], Chaoliang Li[2], Shuang Zhang[2], Weihao Zheng[2], Yantang Chen[2], Jie Ma[2], Xin Zhang[6], Yanming Huang[7] and Shengming Liu[1]

[1] Department of Respiratory Medicine, The First Affiliated Hospital of Jinan University, Guangzhou, Guangdong, China
[2] Department of Critical Care Medicine, Jiangmen Central Hospital, Affiliated Jiangmen Hospital of Sun Yat-sen University, Jiangmen, Guangdong, China
[3] BGI PathoGenesis Pharmaceutical Technology Co., Ltd, BGI-Shenzhen, Shenzhen, Guangdong, China
[4] BGI Wuhan Biotechnology, BGI-Shenzhen, Wuhan, Hubei, China
[5] Department of Neurology, Jiangmen Central Hospital, Affiliated Jiangmen Hospital of Sun Yat-sen University, Jiangmen, Guangdong, China
[6] Clinical Experimental Center, Jiangmen Central Hospital, Affiliated Jiangmen Hospital of Sun Yat-sen University, Jiangmen, Guangdong, China
[7] Department of Respiration Medicine, Jiangmen Central Hospital, Affiliated Jiangmen Hospital of Sun Yat-sen University, Jiangmen, Guangdong, China

Corresponding author
Shengming Liu, smliu01@163.com, tlsm@jnu.edu.cn

## ABSTRACT

**Background:** Metagenome next-generation sequencing (mNGS) is a valuable diagnostic tool that can be used for the identification of early pathogens of acute respiratory distress syndrome (ARDS) in severe pneumonia. Little is known about the use of this technology in clinical application and the evaluation of the prognostic value of ARDS.

**Methods:** We performed a retrospective cohort study of patients with ARDS caused by severe pneumonia. Samples were collected from patients in the intensive care unit (ICU) of Jiangmen Central Hospital from January 2018 to August 2019. The no-next generation sequencing (NGS) group was composed of patients given conventional microbiological tests to examine sputum, blood, or bronchoalveolar lavage fluid. The NGS group was composed of patients tested using mNGS and conventional microbiological tests. We evaluated the etiological diagnostic effect and clinical prognostic value of mNGS in patients with ARDS caused by severe pneumonia.

**Results:** The overall positive rate (91.1%) detected by the mNGS method was significantly higher than that of the culture method (62.2%, $P = 0.001$), and antibody plus polymerase chain reaction (28.9%, $P < 0.001$). Following adjustment of the treatment plan based on microbial testing results, the Acute Physiology and Chronic Health Evaluation-II (APACHE II) score of the NGS group was lower than that of the no-NGS group 7 days after treatment ($P < 0.05$). The 28-day mortality rate of the NGS group was significantly lower than that of the no-NGS group ($P < 0.05$). Longer ICU stay, higher APACHE II score and sequential organ failure assessment
score were risk factors for the death of ARDS, and adjusting the medication regimen based on mNGS results was a protective factor. The detection of mNGS can significantly shorten the ICU stay of immunosuppressed patients ($P < 0.01$), shorten the ventilation time ($P < 0.01$), and reduce the ICU hospitalization cost ($P < 0.05$).

**Conclusions:** Metagenome next-generation sequencing is a valuable tool to determine the etiological value of ARDS caused by severe pneumonia to improve diagnostic accuracy and prognosis for this disease. For immunosuppressed patients, mNGS technology can be used in the early stage to provide more diagnostic evidence and guide medications.

## INTRODUCTION

Acute respiratory distress syndrome (ARDS) is typically caused by infections, such as pneumonia (*Saguil & Fargo, 2012*). Failure of timely and effective treatment will lead to multiple organ failure and death. Approximately 31% (*Griffiths et al., 2019*) of patients with ARDS are admitted to the intensive care unit (ICU), with a mortality rate of 19.7–57.7% (*Bein et al., 2016*; *Bellani et al., 2016*; *Griffiths et al., 2019*). ARDS survivors are at greater risk of cognitive decline, depression, post-traumatic stress disorder, and persistent skeletal muscle weakness (*Herridge et al., 2016*; *Herridge et al., 2011*), bringing a great economic burden to families and society. Early pathogen identification and clinical intervention are critical for ARDS patients to reduce mortality and improve prognosis (*Lee, 2017*).

Conventional microbiological testing includes bacterial/fungal culture, polymerase chain reaction (PCR) nucleic acid hybridization, and serological antibody testing. The turn-around time of bacterial/fungal cultures is long (3–5 days), and the positive rate is low (*Miao et al., 2018*). PCR nucleic acid hybridization requires pre-screening of microbial pathogens and designing specific primers/probes, but detection types are limited (*Spackman et al., 2002*). There is a window period that cannot be accurately identified by the serological antibody detection (*Rajapaksha et al., 2019*). Metagenome next-generation sequencing (mNGS) was first used to diagnose a central nervous system (CNS) infection of *Leptospira* in 2014 (*Wilson et al., 2014*). This emerging diagnostic technology can quickly detect all nucleic acids in specimens of different sample types in one test, including blood, respiratory tract, CNS, and focal tissue (*Guan et al., 2016*; *Guo et al., 2019*; *Long et al., 2016*; *Miao et al., 2018*). mNGS technology has been successfully used clinically for rapid identification of pathogens in ARDS patients with pneumonia (*Fischer et al., 2014*) and can be used in clinical diagnosis and drug decision-making of severe pneumonia (*Yang et al., 2019*).

Different physiological indicators are crucial to the development and prognosis of ARDS in patients. Reduction of platelet count following ICU admission, age, body mass

index, immunocompromised status, prone positioning, days of mechanical ventilation, disease score, elevated cardiac troponin T, extent of endothelial injury, low $PaO_2/FiO_2$ ratio, and different clinical intervention treatment options (*Chen & Ware, 2015*) affects the prognosis of patients with ARDS. Prior analysis of the prognosis of patients with ARDS using multiple Cox regression models found that late-onset moderate to severe ARDS was associated with adverse outcomes (*Zhang et al., 2017*). However, the effect of mNGS technology on the prognosis of ARDS patients is unknown.

Currently, the clinical application of mNGS in ARDS appears predominantly in case reports or small-scale cohort studies. There is an urgent need to review the practical application of mNGS technology in ARDS patients, and assess its prognostic value. Thus, this study summarizes clinical information via retrospective analysis, and evaluates the clinical prognosis of ARDS by mNGS technology application.

## MATERIALS AND METHODS

### Ethical approval and consent

The protocol used in this retrospective study was reviewed and approved by the Ethical Review Committee of Jiangmen Central Hospital (No: 2019-15). Patient's informed consent was obtained from patients or their next of kin.

### Study participants

A retrospective analysis was conducted on all ARDS cases resulting from severe pneumonia in patients 18 years and older, admitted to the ICU at Jiangmen Central Hospital from January 2018 to August 2019. For our study, ARDS was diagnosed according to the 2012 Berlin definition of the disease (*ARDS Definition Task Force, 2012*). Patients were excluded from the study if their ARDS was not caused by severe pneumonia or if they did not follow through with their treatment for any reason.

All patients were endotracheally intubated, mechanically ventilated, and underwent a fiberoptic bronchoscopy to obtain clinical specimens for microbial testing. Patients were included in the next generation sequencing (NGS) group when informed consent was provided for testing; those who were not tested by mNGS were grouped into the no-NGS group. Owing to the cost of mNGS, only DNA sequencing was performed. Samples of bronchoalveolar lavage fluid (BALF) were acquired from patients in the NGS group and sent for pathogen testing at BGI Clinical Laboratories (Shenzhen) Co., Ltd. Once the laboratory received the samples, nucleic acid extraction, library construction, high-throughput sequencing, bioinformatics analysis, and pathogen data interpretation were performed according to previous studies (*Miao et al., 2018*).

### Microbiological testing

Both groups were tested using the same conventional method (routine culture + serum antibody + PCR). The NGS group used mNGS + routine culture + serum antibody + PCR, while the no-NGS group used routine culture + serum antibody + PCR. Pathogenic microbes that cause severe pneumonia are typically bacteria, fungi, or viruses. Restricted by inspection conditions of the hospital, serum antibody and PCR nucleic acid detection

could only detect some special pathogens and viruses that were clinically difficult to culture, as a supplement to routine culture. The serum antibody included *Mycoplasma pneumoniae, Chlamydia pneumoniae*, Coxsackie virus, cytomegalovirus, influenza A, influenza B, respiratory syncytial virus, and parainfluenza virus. PCR nucleic acids included *Legionella pneumophila, Mycoplasma pneumoniae, C. pneumoniae, Mycobacterium tuberculosis*, influenza A, and influenza B.

### Clinical treatment

All patients underwent empirical antimicrobial treatment according to the Chinese Adult Community-Acquired Pneumonia Diagnosis and Treatment Guide (*Cao et al., 2018*) and the Chinese Adult Hospital-Acquired Pneumonia and Ventilator-associated Pneumonia Diagnosis Guide (*Department of Infectious Diseases, Chinese Medical Association Respiratory Branch, 2018*), combined with respiratory infection indicators and imaging. All patients were treated with mechanical ventilation according to the ARDS ventilation guidelines (*Bein et al., 2016*; *Griffiths et al., 2019*). The no-NGS patients were treated with an antimicrobial regimen based on the results of conventional microbiological tests. The antimicrobial regimen of NGS patients were adjusted case-by-case according to mNGS results.

### Information collection and analysis

Patient data included age, gender, basic disease, laboratory test results before treatment, ventilator parameters, conventional microbiological tests, serum biomarkers, ICU special treatment data, APACHE II, and SOFA scores. Data were collected and compared between the two groups. The primary outcome was measured by a 28-day all-cause mortality. Secondary outcomes were measured as the length of stay in the ICU, duration of mechanical ventilation, duration of extracorporeal membrane oxygenation (ECMO), duration of prone ventilation positioning, and ICU treatment costs. Patients that showed signs of immunosuppression were selected from both groups and their prognosis compared using the same aforementioned outcomes. Cox regression analysis was conducted to analyze risk factors for ARDS prognosis. The mNGS results were compared with those of conventional microbiological tests in the NGS group.

### Statistical analysis

The *t*-test was used to determine normal distribution and uniformity of variance. The Wilcoxon rank test was used to calculate variance of measured data that were not normally distributed or had variance homogeneity. The chi-square test was used to calculate the difference between both groups. All statistical analyses were conducted using GraphPad 5.0 and R3.4.4 software. $P < 0.05$ was considered statistically significant.

## RESULTS

### Sample and patient characteristics

A total of 105 patients with ARDS caused by severe pneumonia were screened in this study and 10 patients were excluded based on exclusion criteria. Fourty two patients were placed

**Table 1 Patient characteristics and baseline of two groups.**

|  | NGS (*n* = 42) | no-NGS (*n* = 53) | *P*-value |
|---|---|---|---|
| Age (yr) | | | |
| ≥ 60, *n* (%) | 21 (50.0) | 33 (62.3) | 0.231 |
| < 60, *n* (%) | 21 (50.0) | 20 (37.7) | |
| Gender | | | |
| Male, *n* (%) | 31 (73.8) | 38 (71.7) | 0.819 |
| Female, *n* (%) | 11 (26.2) | 15 (28.3) | |
| Basis disease | | | |
| Hypertension, *n* (%) | 13 (31.0) | 17 (32.1) | 0.907 |
| Coronary heart disease, *n* (%) | 3 (7.1) | 5 (9.4) | 0.690 |
| COPD, *n* (%) | 10 (23.8) | 17 (32.1) | 0.375 |
| Chronic nephrosis, *n* (%) | 7 (16.7) | 6 (11.3) | 0.452 |
| Diabetes, *n* (%) | 5 (11.9) | 9 (17.0) | 0.488 |
| Immunosuppression, *n* (%) | 8 (19.0) | 13 (24.5) | 0.523 |
| Tumor, *n* (%) | 10 (23.8) | 11 (20.8) | 0.722 |
| Smoking, *n* (%) | 20 (47.6) | 17 (32.1) | 0.123 |
| Drinking, *n* (%) | 4 (9.5) | 5 (9.4) | 0.988 |

Notes:
COPD, chronic obstructive pulmonary disease.
There were no any differences in age, sex ratio, basis disease between two groups ($P > 0.05$).
The chi-square test was utilized to calculate the difference between the two groups.
$P < 0.05$ was considered statistically significant.

into the NGS group and 53 patients in the no-NGS group. Three patients in the NGS group had two mNGS tests performed and a total of 45 BALF samples were sent for mNGS.

Patient demographics, characteristic baselines, and ICU special treatments in the NGS and no-NGS groups were shown in Tables 1–3, respectively. There were no significant differences in age, gender, basic disease, laboratory test results before treatment, ventilator parameters, APACHE II and SOFA scores before treatment, and incidences of special treatment in the ICU between both groups ($P > 0.05$).

## Comparison of outcomes between NGS and no-NGS groups

There was a significant difference in the 28-day all-cause mortality between both groups ($P = 0.006$) (Table 4). The 28-day survival was significantly higher in the NGS group than in the no-NGS group (Hazard Ratio = 2.41, 95% CI: 1.21–4.17, $P = 0.01$) (Fig. 1). There was no significant difference in the length of stay in the ICU, duration of mechanical ventilation, ECMO, prone position ventilation, or the cost of the ICU stay between both groups ($P > 0.05$) (Table 4).

## Prognosis of ARDS patients

Cox univariate analysis was performed on all factors and Cox multivariate analysis was performed with variates which were $P < 0.2$ of the Cox univariate analysis (Table S1). The NGS or no-NGS group, length of stay in ICU, and APACHE II and SOFA scores

**Table 2 Laboratory examination before treatment, Ventilator parameters, APACHE II score and SOFA score before treatment of two groups.**

|  | NGS ($n = 42$) | no-NGS ($n = 53$) | *P*-value |
|---|---|---|---|
| Laboratory examination before treatment |  |  |  |
| PCT (ug/L) | 1.3 (0.5, 8.4) | 2.5 (0.3, 10.6) | 0.516 |
| WBC ($10^9$/L) | 10.5 (6.4, 15.4) | 13.1 (7.5, 15.5) | 0.189 |
| Hb (g/L) | 109 (85, 130) | 105 (84, 129) | 0.932 |
| PLT ($10^9$/L) | 159 (84, 205) | 154 (112, 197) | 0.780 |
| Scr (μmol/L) | 78 (64, 201) | 97 (64, 121) | 0.515 |
| T.Bil (mmol/L) | 11.8 (5.2, 17.2) | 14.4 (7.8, 21.1) | 0.071 |
| ALT (IU/L) | 28 (20, 47) | 27 (20, 45) | 0.612 |
| Alb (g/L) | 28.0 (23.6, 31.6) | 28.2 (24.8, 32.6) | 0.880 |
| APTT (sec) | 35.6 (31.0, 44.7) | 34.7 (26.4, 48.1) | 0.614 |
| NT-proBNP (pg/ml) | 652 (236, 2747) | 656 (311, 2066) | 0.482 |
| Lac (mmol/L) | 1.6 (1.4, 2.9) | 1.7 (1.2, 2.5) | 0.763 |
| Ventilator parameters |  |  |  |
| $FiO_2$ | 0.8 (0.6, 1.0) | 0.6 (0.5, 0.8) | 0.992 |
| Peep | 10 (8, 15) | 8 (6, 12) | 0.272 |
| OI | 124 (76, 177) | 156 (108, 194) | 0.996 |
| APACHE II score before treatment | 22 (18, 26) | 21 (17, 26) | 0.500 |
| SOFA score before treatment | 7 (5, 8) | 7 (4, 8) | 0.875 |

Notes:

PCT, Procalcitonin; WBC, White blood cell; Hb, Hemoglobin; PLT, Platelet count; Scr, Serum creatinine; T.Bil, Total bilirubin; ALT, Alanine aminotransferase; Alb, Albumin; APTT, Activated partial thromboplastin time; NT-proBNP, N-terminal Pro-Brain Natriuretic Peptide; Lac, Lactate; $FiO_2$, Fraction of inspiration $O_2$; Peep, positive end-expiratory pressure; OI, Oxygenation Index; APACHE-II, Acute physiology and chronic health evaluation-II; SOFA, Sequential organ failure assessment.

There were no any differences in laboratory examination, ventilator parameters, APACHE II score and SOFA score before treatment between two groups ($P > 0.05$).

The measured data of patients' physiological indicators in the above table were shown by median (interquartile range).

$P < 0.05$ was considered statistically significant.

**Table 3 ICU special treatment of two groups.**

|  | NGS ($n = 42$) | no-NGS ($n = 53$) | *P*-value |
|---|---|---|---|
| Use of vasoactive agent, *n* (%) | 24 (57.1) | 30 (56.6) | 0.958 |
| CRRT, *n* (%) | 9 (21.4) | 7 (13.2) | 0.288 |
| ECMO, *n* (%) | 6 (14.3) | 3 (5.7) | 0.177 |
| Prone positioning, *n* (%) | 10 (23.8) | 11 (20.8) | 0.722 |

Notes:

ICU, intensive care unit; CRRT, continuous renal replacement therapy; ECMO, extracorporeal membrane oxygenation.

There were no any differences in ICU special treatment between two groups ($P > 0.05$).

The chi-square test was utilized to calculate the difference between the two groups.

$P < 0.05$ was considered statistically significant.

before treatment were risk factors in patients with ARDS caused by severe pneumonia. The NGS group patients had a better prognosis than that of the no-NGS group patients ($P = 0.005$). A shorter stay in the ICU ($P = 0.037$), and lower APACHE II ($P = 0.016$) and SOFA scores before treatment ($P = 0.003$) had a better prognosis (Table 5).

**Table 4 Comparison of outcomes between NGS and no-NGS groups.** The primary outcome: There was a significant difference in 28-day all-cause mortality between the two groups ($P = 0.006$). The secondary outcome: There was no significant difference in the length of stay in the ICU, the duration of mechanical ventilation, ECMO, prone position ventilation, or the cost of the ICU stay between the two groups ($P > 0.05$). The chi-square test was utilized to calculate the difference between the two groups in the primary outcome. The $t$-test was utilized to calculate the difference between the two groups in the secondary outcome. The measured data of patients' outcomes in the above table were shown by median (interquartile range).

| | NGS ($n = 42$) | no-NGS ($n = 53$) | P-value |
|---|---|---|---|
| The primary outcome | | | |
| 28-day all-cause mortality | 9 (21.4%) | 26 (49.1%) | 0.006* |
| The secondary outcomes | | | |
| Length of stay in ICU (days) | 12 (7, 20) | 11 (8, 15) | 0.719 |
| Duration of mechanical ventilation (h) | 240 (144, 353) | 216 (134, 311) | 0.810 |
| Duration of ECMO (days) | 15 (11, 18) | 10 (10, 23) | 0.500 |
| Duration of prone position ventilation (h) | 89 (63, 117) | 96 (71, 121) | 0.345 |
| Cost in ICU (thousand CNY) | 82.3 (55.1, 211.1) | 98.9 (68.9, 141.1) | 0.297 |

**Note:**
* $P < 0.05$ was considered statistically significant.

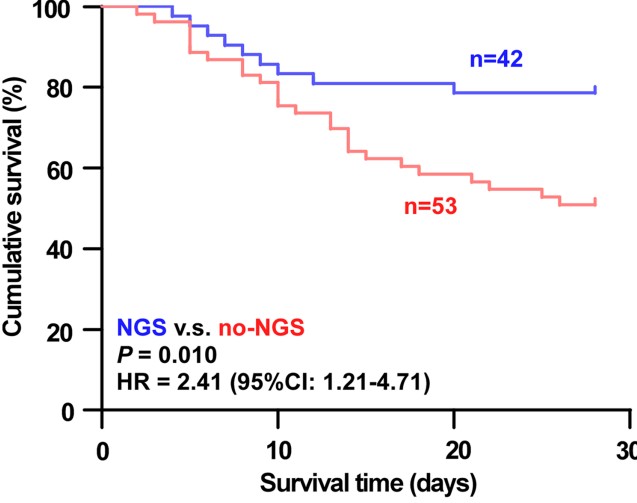

**Figure 1 Analysis of 28-day survival curves of patients in the NGS group and the no-NGS group.** The 28-day survival was significantly higher in the NGS group than in the no-NGS group (HR = 2.41, 95% CI: 1.21–4.17, $P = 0.01$).

## Comparison of mNGS results and culture results in the NGS group

The current research showed that the mNGS test can detect more pathogens than the culture method. We analyzed the consistency of pathogens identified by both techniques. The test results were considered to be consistent when the pathogens identified by mNGS were the same as the pathogens obtained from culture. The test results were also considered consistent if mNGS identified more pathogens than the culture method. The result was partially consistent when pathogens identified by both methods were partially congruent. The results were considered inconsistent when pathogens identified by

**Table 5 Cox multivariate analysis of prognosis of patients with ARDS.** The NGS group had a better prognosis than no-NGS group ($P = 0.005$). Those with a shorter stay in the ICU ($P = 0.037$), and lower APACHE II before treatment ($P = 0.016$) and SOFA scores before treatment ($P = 0.003$) had a better prognosis.

| | HR | Lower .95 | Upper .95 | *P*-value |
|---|---|---|---|---|
| mNGS (yes/no) | 0.263 | 0.105 | 0.663 | 0.005* |
| Age (years) | 1.013 | 0.988 | 1.038 | 0.322 |
| Length of stay in ICU (days) | 0.888 | 0.794 | 0.993 | 0.037* |
| APACHE II score before treatment | 1.112 | 1.020 | 1.212 | 0.016* |
| SOFA score before treatment | 1.339 | 1.105 | 1.622 | 0.003* |
| Coronary heart disease (yes/no) | 1.660 | 0.556 | 4.958 | 0.364 |
| Bronchiectasis (yes/no) | 1.128 | 0.331 | 3.843 | 0.848 |
| Diabetes (yes/no) | 0.324 | 0.088 | 1.195 | 0.091 |
| Hb (g/L) | 0.993 | 0.980 | 1.006 | 0.284 |
| T.Bil (mmol/L) | 0.999 | 0.987 | 1.012 | 0.882 |
| Be | 1.063 | 0.996 | 1.133 | 0.064 |
| Use of vasoactive agent (yes/no) | 1.443 | 0.587 | 3.548 | 0.424 |
| ECMO (yes/no) | 1.212 | 0.067 | 21.764 | 0.896 |
| Cost in ICU (CNY) | 1.000 | 1.000 | 1.000 | 0.477 |

**Note:**
\* $P < 0.05$ was considered statistically significant.

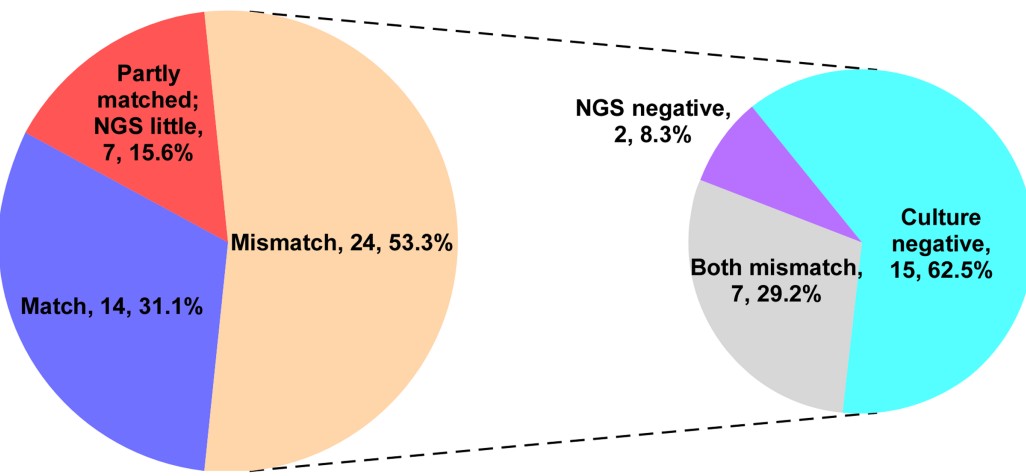

**Figure 2 The consistent analysis comparing culture and mNGS pathogen detection in the NGS group.** Identified pathogens (31.1%) in the NGS group were consistent, 15.6% were partially consistent, and 53.3% were completely inconsistent. In the inconsistent ones, 62.5% were negative for the culture method, while 8.3% were negative for mNGS, and 29.2% were mismatched.

both methods varied completely. Identified pathogens (31.1%) in the NGS group were consistent, 15.6% were partially consistent, and 53.3% were completely inconsistent. In the inconsistent ones, 62.5% were negative for the culture method, while 8.3% were negative for mNGS, and 29.2% were mismatched (Fig. 2).

**Table 6 Comparison of metagenomic NGS results and conventional microbiological tests.** The positive rate of mNGS virus detection was lower than that of serum antibody detection plus PCR (6.7% vs. 26.7%, $P = 0.021$). mNGS was significantly better at detecting bacteria than serological antibody testing plus PCR (24.4% vs. 0%, $P = 0.001$). Further, mNGS was able to detect specific pathogens better than the culture method (22.2% vs. 0%, $P = 0.001$) and serological antibody testing plus PCR (22.2% vs. 2.2%, $P = 0.007$). Additionally, mNGS was significantly better at the identification of co-infections than serological antibody tests plus PCR (26.7% vs. 0%, $P < 0.001$). Finally, mNGS proved to be significantly better at identifying pathogens than the culture method (91.1% vs. 62.2%, $P = 0.001$) and serological antibody testing plus PCR (91.1% vs. 28.9%, $P < 0.001$).

| | Method A ($n = 45$) | Method B ($n = 45$) | Method C ($n = 45$) | P-value, A vs. B | P-value, A vs. C |
|---|---|---|---|---|---|
| Only virus, $n$ (%) | 3 (6.7) | 0 (0.0) | 12 (26.7) | 0.24 | 0.021* |
| Only bacterial, $n$ (%) | 11 (24.4) | 15 (33.3) | 0 (0.0) | 0.486 | 0.001* |
| Only fungus, $n$ (%) | 5 (11.1) | 5 (50.0) | 0 (0.0) | 1 | 0.056 |
| Special pathogen, $n$ (%) | 10 (22.2) | 0 (0.0) | 1 (2.2) | 0.001* | 0.007* |
| Co-infection, $n$ (%) | 12 (26.7) | 8 (17.8) | 0 (0.0) | 0.311 | <0.001* |
| Overall positive, $n$ (%) | 41 (91.1) | 28 (62.2) | 13 (28.9) | 0.001* | <0.001* |

Notes:
Method A: mNGS; Method B: Culture; Method C: Serological antibody test plus PCR.
The Chi-square test was utilized to calculate the difference between the two groups.
* $P < 0.05$ was considered statistically significant.

## Comparison metagenomic of NGS results and conventional microbiological tests

Some special pathogens were difficult to obtain via culture. Therefore, *Legionella*, *Tuberculosis*, *Mycoplasma/Chlamydia*, parasites, K. spores, *etc.* were defined as such. Severe pneumonia is not caused by a single pathogen and is typically accompanied by co-infections. A co-infection is defined as a non-single pathogenic infection, such as bacteria + fungi/bacteria + virus/fungi + virus/bacteria + fungi + virus.

The positive rate of mNGS virus detection was lower than that of serum antibody detection plus PCR (6.7% vs. 26.7%, $P = 0.021$). In this study, mNGS only performed DNA sequencing and could only detect DNA viruses, whereas viruses identified by serological antibody detection and PCR were RNA viruses, such as influenza A and influenza B. mNGS was significantly better at detecting bacteria than serological antibody testing plus PCR (24.4% vs. 0%, $P = 0.001$). Further, mNGS was able to detect specific pathogens better than the culture method (22.2% vs. 0%, $P = 0.001$) and serological antibody testing plus PCR (22.2% vs. 2.2%, $P = 0.007$). Additionally, mNGS was significantly better at the identification of co-infections than serological antibody tests plus PCR (26.7% vs. 0%, $P < 0.001$). Finally, mNGS proved to be significantly better at identifying pathogens than the culture method (91.1% vs. 62.2%, $P = 0.001$) and serological antibody testing plus PCR (91.1% vs. 28.9%, $P < 0.001$) (Table 6).

## Clinical medication guidance between NGS and no-NGS groups

In the NGS group, 30 patients (71.4%) did not cover all the microbial detected by mNGS in the initial empirical antimicrobial treatment. Thus, antimicrobial regimen needs to be modified accordingly based on the mNGS results. In the no-NGS group, empirical

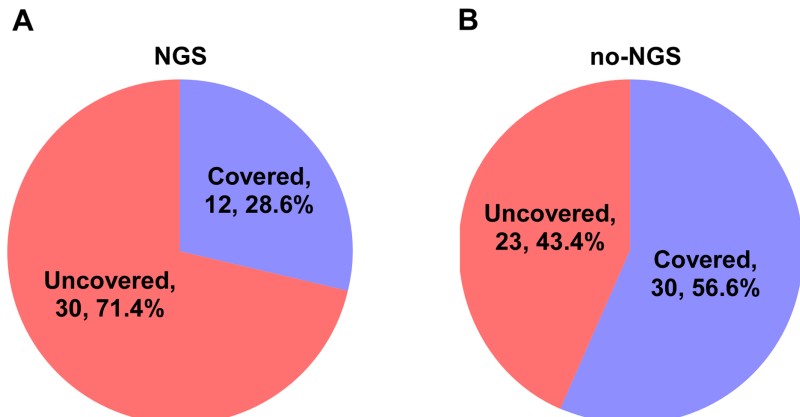

**Figure 3 Coverage spectrum of empirical antimicrobial therapy for pathogen detection results in two groups.** (A) In the NGS group, 30 patients (71.4%) did not cover all the microbial detected by mNGS in the initial empirical antimicrobial treatment. Thus, antimicrobial regimen needs to be modified accordingly based on the mNGS results. (B) In the no-NGS group, empirical antimicrobial treatment that could not cover the detected microbials was found in 23 patients (43.4%), according to the results of traditional microbiological testing, and they were necessary to adjust the anti-infection program.

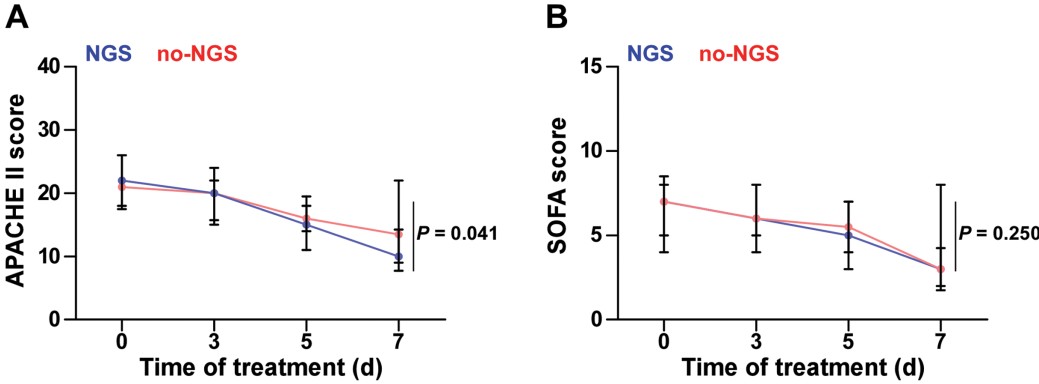

**Figure 4 APACHE II and SOFA scores of the two groups.** (A) The NGS group had a lower APACHE II score than that in the no-NGS group after 7 days of treatment ($P = 0.041$). (B) There was no significant difference in SOFA score during 7 days between two groups ($P > 0.05$).

antimicrobial treatment that could not cover the detected microbials was found in 23 patients (43.4%), according to the results of traditional microbiological testing, and they were necessary to adjust the anti-infection program (Fig. 3). Following adjustment of the anti-infective regimen, we continuously observed APACHE II and SOFA scores for both groups of patients for 7 days and found that the NGS group had a lower APACHE II score than the no-NGS group, 7 days after treatment ($P = 0.041$) (Fig. 4).

## Immunosuppressed patients

Clinical features of immunosuppressed patients were complicated. A total of 21 immunosuppressed patients were enrolled in our study, eight were subjected to mNGS

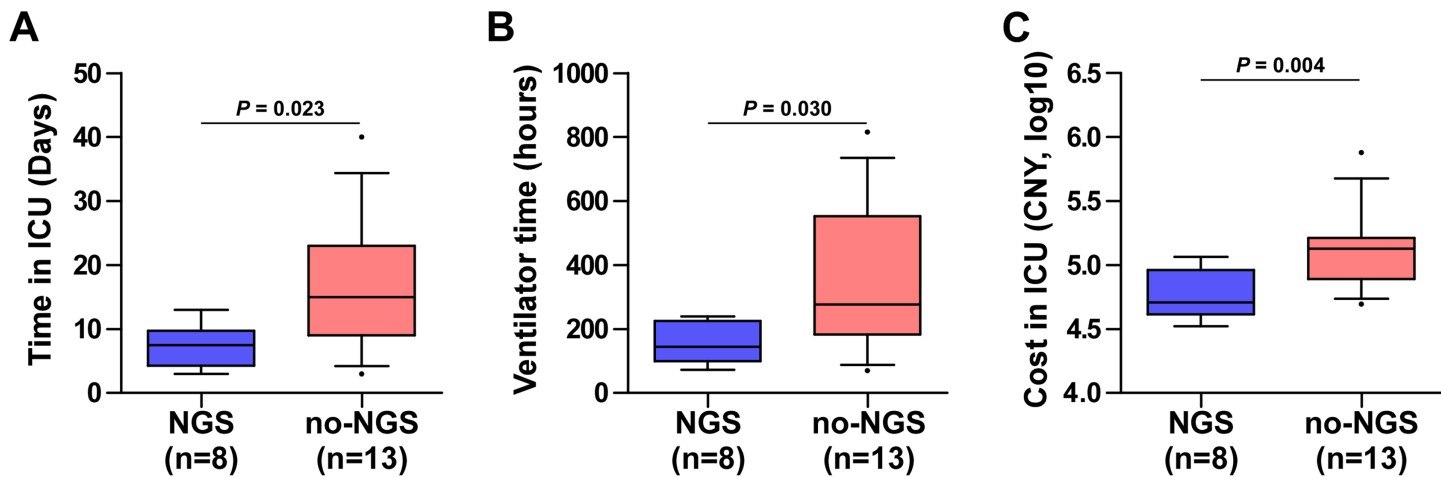

**Figure 5** **Clinical data of 21 immunosuppressed patients with NGS and no-NGS were compared.** The NGS group had shorter length of stay in the ICU (A) ($P$ = 0.023), shorter ventilation time (B) ($P$ = 0.030), and less cost in ICU (C) ($P$ = 0.004) than those in the no-NGS group of immunosuppressed patients.                                                              

pathogen detection, and 13 did not undergo mNGS. Three cultures were positive in the NGS group, consistent with pathogens identified by mNGS, including five *Pneumocystis jirovecii*, one *Rhizopus*, one *Cryptococcus*, and one human herpesvirus; six were co-infections.

In the no-NGS group, nine cases were positive for culture, and two *Stenotrophomonas maltophilia*, two *Acinetobacter baumannii*, one *Staphylococcus aureus*, four *Candida*, and one *Aspergillus* were detected. Four cases had multi-drug resistant bacteria. There was no significant difference in the 28-day all-cause mortality between the two groups (37.5% vs. 53.8%, $P$ = 0.659). However, there were significant differences in the length of stay in the ICU ($P$ = 0.023), duration of mechanical ventilation ($P$ = 0.030), and cost of the stay in the ICU ($P$ = 0.004) between both groups of immunosuppressed patients (Fig. 5).

## DISCUSSION

Acute respiratory distress syndrome caused by severe pneumonia is critical and progresses rapidly. Common microbial infection includes those of bacteria, fungi, and viruses while some are co-infections (*Lee, 2017*). Patients usually require broad-spectrum anti-infection treatment, and then, further adjust to targeted anti-infection treatment based on microbial detection results of. Therefore, it is critical to determine the type of microbial infection for ARDS treatment caused by severe pneumonia.

This study compared the effectiveness of mNGS with traditional microbiological testing methods of the NGS group. Firstly, mNGS was faster, taking an average of 2 days from sending samples to receiving reports, whereas routine culture requires at least 3–5 days. Secondly, the overall positive rate (91.1%) of mNGS was significantly higher than that of culture (62.2%, $P$ = 0.001) and antibody plus PCR (28.9%, $P$ < 0.001). As all patients included were diagnosed with severe pneumonia, the positive rate of mNGS and culture of lower respiratory tract specimens were higher than that of usual detection.

Thirdly, the positive rate of mNGS detection of specific pathogens (22.2%) was higher than that of culture (0%, $P = 0.001$) and antibody plus PCR (2.2%, $P = 0.007$). This conclusion was consistent with a previous study by *Qi et al. (2019)* in that the positive of mNGS was much higher than that of culture, and rare pathogens could be detected. In addition, we analyzed the consistency between mNGS and culture, 31.1% of identified pathogens in the NGS group were consistent, 15.6% were partially consistent, and 53.3% were completely inconsistent. In the inconsistent ones, 62.5% were negative for culture, while only 8.3% were negative for mNGS. The advantages of mNGS detection compared with traditional detection were confirmed.

By comparing the prognosis of patients between the NGS group and the no-NGS group, it was found that the 28-day mortality rate of the NGS group was significantly lower than that of the no-NGS group ($P < 0.05$) (Table 4). There was no difference in ICU hospitalization time, mechanical ventilation time, ECMO time, prone position ventilation time, and ICU treatment costs between the two groups (Table 4). This conclusion was consistent with the study of *Xie et al. (2019)*. They analyzed 178 patients with severe pneumonia and combined mNGS results to guide treatment. The 28-day and 90-day survival rates of severe pneumonia patients were improved. The 90-day survival rate increased from 57.7% to 83.3%.

In this study, clinicians assisted clinical diagnosis through comprehensive microbial testing; the empirical medication of 71.4% of patients in the NGS group did not cover clinically diagnosed microbial infections, whose anti-infection treatment should be adjusted based on mNGS results. In the no-NGS group, 43.4% of patients required adjustment of the empirical anti-infection regimen. Due to faster and more effective adjustment of the anti-infection regimen, it was found that APACHE II scores in the NGS group were lower than those in the no-NGS group 7 days after treatment ($P = 0.041$, Fig. 4). This means that the mNGS test results have a positive effect on clinical medication guidance. Moreover, a multiple Cox regression analysis was conducted for assessment of prognostic factors and found that a longer stay in ICU, high APACHE II score, and high SOFA score were risk factors for ARDS death, and the application of mNGS for clinical pathogen detection was a protective factor. It was shown that the higher the APACHE II and SOFA scores of sepsis patients, the worse the prognosis (*Innocenti et al., 2014*; *Jones, Trzeciak & Kline, 2009*), which is consistent with our results.

In addition, studies have shown that immunosuppressed patients were prone to co-infection. mNGS technology has distinct advantages in detecting co-infection pathogens (*Parize et al., 2017*). In this study, mNGS detected specific pathogens that were difficult to culture in immunosuppressed patients, including *Pneumocystis*, *Rhizopus*, *Cryptococcus*, and viruses. Although the mortality rate of the NGS group was lower than that of the no-NGS group, the difference in the prognostic analysis of immunosuppressed patients was not statistically significant (37.5% vs. 53.8%, $P = 0.659$), and may be related to the small sample size. Moreover, we found that mNGS technology can significantly shorten the length of stay in the ICU of immunosuppressed patients, shorten the ventilation time, and reduce the cost in ICU ($P < 0.05$). From the economics and

clinical prognosis, immunosuppressed patients were more suitable for mNGS technology application in the early clinical stage to assist clinical diagnosis and drug decision-making.

Limitations to the use of mNGS technology exist, despite its widespread use. There is no authoritative guide to the interpretation of the mNGS report. Detection of a broad spectrum of pathogens by mNGS has blunted the diagnosis of pathogenicity resulting in the inability to distinguish between background, colonization and microbial infection, and pollution (*Simner, Miller & Carroll, 2018*). Better technology needs to be developed for mNGS to be used successfully in clinical applications. The use of mNGS in clinical applications will: (1) achieve a faster diagnosis of pathogens and obtain information on drug resistance of related pathogens; (2) identify microbial colonization or infection through monitoring the patient's immune response, which will eventually curb bacterial resistance, achieve a rational application of antibiotics, and ultimately reduce the economic and social burden of infectious diseases; (3) lower the cost of the mNGS test with the development of technology, so that more patients will benefit.

Our research also has certain limitations. Firstly, our mNGS only performed DNA sequencing and did not perform RNA sequencing; therefore, the information of RNA virus and microbial transcriptome alterations were missing, resulting in the positive rate of mNGS virus detection being lower than serum antibody plus PCR (6.7% vs. 26.7%, $P = 0.021$). Secondly, restricted by the inspection conditions of the hospital, PCR detection only included some RNA viruses, such as influenza A and influenza B. Additionally, the prognostic analysis was affected by several clinical factors and sample size of this study was not large, resulting in some data inconsistency. For example, the mortality rate between the two groups was significantly different, but that of the immunosuppressed patients was not. There was no difference in ICU stay, cost, and ventilation time between the two groups, but there was a difference between the two groups of immunosuppressed patients.

## CONCLUSIONS

Metagenome next-generation sequencing technology is valuable for the diagnosis, treatment and prognosis of ARDS caused by severe pneumonia. mNGS technology is superior to conventional microbiological tests for the detection of special pathogens and co-infections. mNGS technology harbors great potential for clinical infection. Further research should include a larger sample size, involving multi-center, prospective, and controlled studies, which will help us better understand the clinical experience summary and prognostic value of mNGS detection in ARDS caused by severe pneumonia.

## ACKNOWLEDGEMENTS

We would like to thank Jacqueline Thai (Peer J) and Editage for English language editing.

### Funding

This study was supported by the grants from the National Natural Science Foundation of China (81802918), National Science and Technology Major Project of China (2018ZX10305409-001-001), the Science and Technology Project of Guangdong Province (2019A1515011565, 2018A030310007), the Science Foundation of Guangdong Province Bureau of Traditional Chinese Medicine (20181273) and the Medical Science Foundation of Jiangmen Central Hospital (J201801). The funders had no role in study design, data collection and analysis, decision to publish, or preparation of the manuscript.

### Grant Disclosures

The following grant information was disclosed by the authors:
National Natural Science Foundation of China: 81802918.
National Science and Technology Major Project of China: 2018ZX10305409-001-001.
Science and Technology Project of Guangdong Province: 2019A1515011565, 2018A030310007.
Science Foundation of Guangdong Province Bureau of Traditional Chinese Medicine: 20181273.
Medical Science Foundation of Jiangmen Central Hospital: J201801.

### Competing Interests

Yan Chen is an employee of BGI Genomics. The authors declare that they have no competing interests.

### Author Contributions

- Peng Zhang performed the experiments, analyzed the data, prepared figures and/or tables, authored or reviewed drafts of the paper, and approved the final draft.
- Yan Chen performed the experiments, analyzed the data, prepared figures and/or tables, authored or reviewed drafts of the paper, and approved the final draft.
- Shuyun Li performed the experiments, authored or reviewed drafts of the paper, and approved the final draft.
- Chaoliang Li performed the experiments, authored or reviewed drafts of the paper, and approved the final draft.
- Shuang Zhang performed the experiments, authored or reviewed drafts of the paper, and approved the final draft.
- Weihao Zheng performed the experiments, authored or reviewed drafts of the paper, and approved the final draft.
- Yantang Chen performed the experiments, authored or reviewed drafts of the paper, and approved the final draft.
- Jie Ma performed the experiments, authored or reviewed drafts of the paper, and approved the final draft.

- Xin Zhang analyzed the data, prepared figures and/or tables, authored or reviewed drafts of the paper, and approved the final draft.
- Yanming Huang conceived and designed the experiments, authored or reviewed drafts of the paper, and approved the final draft.
- Shengming Liu conceived and designed the experiments, authored or reviewed drafts of the paper, and approved the final draft.

## Human Ethics

The following information was supplied relating to ethical approvals (i.e., approving body and any reference numbers):

The protocols used in this retrospective study was reviewed and approved by the Ethical Review Committee of Jiangmen Central Hospital (No: 2019-15).

## Data Availability

The datasets generated and analyzed during the current study are available in the Supplemental Files and at Figshare: zhang, peng (2019): raw data.xlsx. figshare. Dataset. DOI 10.6084/m9.figshare.10308617.v3.

## Supplemental Information

Supplemental information for this article can be found online at http://dx.doi.org/10.7717/peerj.9623#supplemental-information.

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
