# Peer review of "Metagenomic next-generation sequencing for the clinical diagnosis and prognosis of acute respiratory distress syndrome caused by severe pneumonia: a retrospective study"

_PeerJ, doi:10.7717/peerj.9623_

## Round 0.1 · original submission · Minor Revisions

Four independent reviewers have assessed your manuscript and all concur in suggesting some changes. I encourage you to consider all of them, as there is some overlap, and these surely will help improve the clarity of presentation of your findings.

Reviewer 1 ·

Basic reporting

no comment

Experimental design

no comment

Validity of the findings

no comment

Additional comments

In this manuscript, Zhang et al investigate the value of metagenomic next-generation sequencing (mNGS) in acute respiratory distress syndrome (ARDS) caused by severe pneumonia. The present study included 42 patients using mNGS method (NGS group) and 53 patients undergoing conventional microbiological tests (no-NGS group) with ARDS caused by severe pneumonia. They found that the mortality was significantly lower and the 28-day survival rate was higher in NGS group than that of in the no-NGS group. Furthermore, mNGS was incredibly reliable for detecting special pathogens and patients with co-infections compared with conventional microbiological tests. The authors demonstrated that mNGS provide more evidence for diagnosis and guidance for treatment. In general, the study is well designed and the results are convincing. The manuscript is well written and the data are presented in an appropriate way. So, I think it is suitable for publication with some modification in writing.

·

Basic reporting

no comment

Experimental design

no comment

Validity of the findings

no comment

Additional comments

In this study entitled "Value of metagenomic next-generation sequencing for the clinical diagnosis and prognosis of acute respiratory distress syndrome caused by severe pneumonia", the authors investigated that mNGS is a valuable diagnostic tool to detect a broad range of pathogens, especially in detecting co-infection for patients with ARDS caused by severe pneumonia in ICU. The whole study might provide some important significance to clinical diagnosis and even treatment for patients with ARDS caused by severe pneumonia. However, I have a few specific comments that should be addressed to improve the manuscript:
1. Line 248, “The positive rate virus and bacterial identification by mNGS was 3 times greater than routine methods.” This is obviously confused from the results represented in Table 6. The author should illustrate it.
2. The result of this article showed that mNGS could detect more microorganisms compared with other methods, which may be the reason of better prognosis. The author considered a test is more reliable and better when the positive rate is higher. However, a higher positive rate does not always mean better. Could the author provide some evidence or reference about false positive, false negative on sensitivity and specificity for mNGS as well as for other methods?
3. Writing of this article should be further modified.

Reviewer 3 ·

Basic reporting

no comment

Experimental design

well done

Validity of the findings

well

Additional comments

General comments:
This research article entitled “Value of metagenomic next-generation sequencing for the clinical diagnosis and prognosis of acute respiratory distress syndrome caused by severe pneumonia”, the authors by Peng Zhang, Yan Chen, Shuyun Li, Chaoliang Li, Shuang Zhang, Weihao Zheng, Yantang Chen, Jie Ma, Xin Zhang, Yanming Huang and Shengming Liu, extensively and detailed investigate the diagnostic and prognostic values of metagenomics next generation sequencing (mNGS) in evere acute respiratory distress syndrome (ARDS) characterized by alveolar collapse, pulmonary edema and elevated epithelial permeability, finally resulting in hypoxia and subsequent deterioration of pulmonary function. Although great advance in the diagnosis and therapy of ARDS over the last decades, the motality in ARDS patients remain higher than 30%. Therefore, accumulating attention has been made to identify the early diagnosis and targeted medicine against ARDS. mNGS is a kind of novel detection technique, which study the DNA and RNA information of microorganism in specific scenario. Importantly, this technique is extensively reported to be applied in detecting blood flow, respiratory tract, CNS disease and abscess. In the current study, the author reported, for the first time, the diagnostic and prognostic values of metagenomics next generation sequencing (mNGS) in evere acute respiratory distress syndrome (ARDS) and concluded that mNGS can be used an important avenue to improve the precision for diagnosis and prognosis of severe pneumonia-induced ARDS, which is beneficial for clinical guide for the treatment of ARDS. Overall, this study is well design and performed, and the manuscript is well written. However, several issues need to be addressed before consideration of acceptance of publication:

Major concerns:
1. The main results of the current study were not clearly shown in the Abstract. Please re-write the corresponding contents.

2. What did “positive rate” stand for in the Abstract? Furthermore, the positive rate in NGS group seemed not to be higher than the sum of other two values. Please clarify.

3. In the manuscript title, the prognosis was mentioned. However, there was deficient of relevant contents in the main text. Please correct.

4. In the introduction section: omission of several important literatures regarding the application of mNGS in the diagnosis of ARDS was a drawback of the current MS. Please correct.

5. In the introduction section: the introduction about the therapy of ADRS could be shortened, cuz the topic of this study focused on the the diagnostic and prognostic values of metagenomics next generation sequencing (mNGS) in evere acute respiratory distress syndrome (ARDS).

6. In the introduction section: a summary of this study should be provided, including the study aim and contents.

7. In methods: what is the metagenomics? Please provide the relevant method or reference.

8. In methods: Please provide the relevant method or reference about the BALF.

9. In resuts: the information in the figure was not matched with the description in the main text, for example, Fig. 1and Figure 2. In table 4, the statictical method used for primary outcome was chi-square test,but the secondary outcome was T-test, not the chi-square test. Please correct.

10. The introduction or discussion should not be mentioned in the Results section, for example when comparing the results of mNGS and culture.

11. In the discussion: 1. What the cause for the differential motality of ARDS patients in ICU? They have the similar ICU duration, expense and ventilation duration. 2. In the immunocompromised patients, They have the different ICU duration, expense and ventilation duration, but did not have statistical significance between them. Why?

12. In the discussion: the paragraph of limitation regarding the mNGS should be shortened.

Minor comments:
1. There were multiple language issues existing in the current study, including:
(1)Line 47: a negative sputum culture,“a” should be deleted
(2)Line 52: mNGS is valuable tool,chaned to“mNGS is a valuable tool”;
(3)Line 74: is chaned to are;
(4)Line 83: next chaned to next-;
(5)Line 117,118: Community and Hospital chaned to Community-和Hospital- ;
(6)Line 133: from chaned to of;
(7)Line 137: the second was chaned to were;
(8)Line 193: than no-NGS group chaned to than the no-NGS group;
(9)Line 243: a virus infection, “a” should be deleted

2. In figure 3: empirical antibiotics should be indicated in the image.

3. In figure 4: noNGS in figure, but no-NGS in the main text. Please keep the form in consistence.

·

Basic reporting

The study is the description of use for mNGS identification Data (etiological agents) as of value for accurate cause-effect establishment of ARDS. Paper states that mNGS information is valuable for subsequent disease management including pathogen-specific treatment and prognosis. Amid respiratory diseases, severe pneumonia can get complicated to ARDS depending on a number of factors from Host and from pathogens, and the study analyzed if mNGS data could make a difference in the management of these kind of illness.

Text of the manuscript is in need for some checks for proper English grammar and syntaxes. Please refer to annotated copy of the manuscript. i.e. Line 40: To use immunosuppressed patients (past tense), instead of immunosuppressive. See also lines 60, 95, etc.

-Lack of definition for some abbreviated terms (i.e. line 152. HR = Hazard Ratio).

-Lack of reference for valuable information (line 129: Cox regression analysis).

-Lines 59-67 could be re-phrased to better describe the context of the study. Please refer to annotated copy for details.

-Lines 76-77. It is widely accepted that pathogen isolation is critically dependent on pathogen load in the clinical specimen. Authors stated the opposite. Please support with references.

-Hard to find agreement if BALF clinical specimens are tested for mNGS, while for microbiological method the specimen is Sputum. Specimen are expected to be exactly the same for side-by-side comparison. Otherwise, some annotations are needed in order to better compare differences in specimens.

-Sensitivity of Sputum culture method is higher than ¨Serological antibody test plus PCR¨ method. This is hard to explain. Please consider contamination of specimens used for culturing during handling and testing…please see next comment:

-Line 78: Expected is 15-25 % detection rate using Sputum culture. So, why in this study the detection rate was 62,2 % ??. This point has to be explained in detail. There is no antecedent of such high detection rates.

-In case.control studies the methods have to be exactly the same and tested identically, otherwise wrong conclusions coul be made. Were serological-PCR methods tested for all potential bacteria and other microbes as causes of ARDS ?

-Line 80: Please rephrase: pathogenic MICROBES…or delete pathogenic, or use a different wording to avoid confusions.

-Lines 81-82: Please provide reference. Respiratory pathogens use to be the main focus of development for diagnostic routines using NAT-based reagents, and serological approaches as well. Please better support statement.

-Line 105. Please rephrase: Clinical specimens FOR MICROBIAL CULTURING TESTING.

-Line 120: Anti MICROBIAL (including anti-fungi )

-Revise match between references and cites in text. Example citation of ref 18, line 244. It does not match.

Raw data findings:
-Data from subject80 (P76) must be corrected in excel table, column ARDS degree. Value of OI is higher than 200, thus is mild, not moderated as stated in the database.

-Column ¨Serological antibody test plus PCR¨ only shows a few results (8) in group nonNGS. It would be convenient to disclose (materials and methods) which specific PCR tests were performed in order to better appreciate this specific result.

-Column ¨Serological antibody test plus PCR¨ shows 13 agents detected in group NGS. 12 results are Influenza, none detected in NGS group ?). Authors disclosed that the NGS analysis were focused on DNA viruses, however there are a number of RNA viruses causing respiratory illness, this specific topic has to be better clarified in experimental design.

-The only one results matching between ¨NGS¨ and ¨Serological antibody test plus PCR¨ is M. pneumoniae (subject 24, P22). It would be useful to detail the protocols used for the molecular/PCR diagnostic routines.

Experimental design

-Hard to find agreement if BALF clinical specimens are tested for mNGS, while for microbiological method the specimen is Sputum. Specimen are expected to be exactly the same for side-by-side comparison. Otherwise, some annotations are needed in order to better compare differences in specimens.

-Sensitivity of Sputum culture method is higher than ¨Serological antibody test plus PCR¨ method. This is hard to explain. Please consider contamination of specimens used for culturing during handling and testing…please see next comment:

Line 78: Expected is 15-25 % detection rate using Sputum culture. So, why in this study the detection rate was 62,2 % ??. This point has to be explained in detail. There is no antecedent of such high detection rates.

-In case.control studies the methods have to be exactly the same and tested identically, otherwise wrong conclusions could be made. Were serological-PCR methods tested for all potential bacteria and other microbes as causes of ARDS ?

Validity of the findings

Tables and figures are well designed, and self-contained. Statistical analysis is a plus in achieving conclusions.

Conclusions are properly linked to the results and analysis conducted.

Additional comments

Study is properly planned, rigorously conducted, and convincingly analyzed. A few observations have been included within this review in order to avoid confusions for readers, and to improve the presentation of data. Data will be of value for readers provided than minor adjustments will be done on the manuscript.

---

## Round 0.2 · Minor Revisions

Both reviewers found your manuscript improved. Reviewer 4 had some comments that remains to be addressed.

Reviewer 3 ·

Basic reporting

Well

Experimental design

Well

Validity of the findings

Well

Additional comments

All the concerns have been addressed in the current version of manuscript. Recommend acceptance at this point.

·

Basic reporting

Manuscript PeerJ reviewing Manuscript 45653v2: ¨Value of metagenomic next-generation sequencing for the clinical diagnosis and prognosis of acute respiratory distress syndrome caused by severe pneumonia in retrospective study¨.

The study is the description of use for mNGS identification Data (etiological agents) as of value for accurate cause-effect establishment of ARDS. Paper states that mNGS information is valuable for subsequent disease management including pathogen-specific treatment and prognosis.

Minor comments:

-Text of the manuscript is in need for some checks for proper English grammar and syntaxes. For instance, spaces before and after parenthesis: lines 69, 72, 75, 78, 80, 83, 85, 89, 94, 97, 114, 125, etc.

-Line 81: Please rephrase as ..serological antibody detection (antibody testing)….

-Error entry for references from databases is included in line 140.

-Line 147: Please rephrase as …..adjusted case-by-case according to mNGS results….

-Line 202: Please consider to rephrase.

-Line 223: Please consider to rephrase.

-Lines 224-225: Please consider to rephrase.

-Lines 249250: Please consider to rephrase.

-Line 253: Please complete word, or define abbreviation.

-Line 264: Please consider to rephrase.

-Line 302: Please consider to rephrase.

-Lines 326- 327: Please consider to rephrase.



-Title of tables 4, 5. Please modify title to better represent information contained in the tables.

-Title of table 6. Please amend title to fit English grammar.

-Title for figures 2, 3. Please consider to modify to increase self-explanatory content.

-Figure legends for figures 4, 5. Please consider to rephrase to increase auto containment.

Experimental design

NO COMMENT

Validity of the findings

NO COMMENT

Additional comments

NO COMMENT

---

## Round 0.3 · accepted · Accept

I congratulate you for your work and thank you for taking into account reviewers suggestions.

·

Basic reporting

Manuscript (#45653):
Metagenomic next-generation sequencing for the 1 clinical diagnosis and prognosis of acute respiratory distress syndrome caused by severe pneumonia: a retrospective study. Previous comments/suggestions were properly followed up.

Manuscript will be a valuable contribution to the field. It has many strengths both conceptual, and methodologically.

Experimental design

No comment.

Validity of the findings

No comment.

Additional comments

No comment.